# A Simple and Efficient CRISPR/Cas9 System Using a Ribonucleoprotein Method for *Flammulina filiformis*

**DOI:** 10.3390/jof8101000

**Published:** 2022-09-23

**Authors:** Jianyu Liu, Haiyang Cui, Ruijuan Wang, Zhen Xu, Hailong Yu, Chunyan Song, Huan Lu, Qiaozhen Li, Danrun Xing, Qi Tan, Weiming Sun, Gen Zou, Xiaodong Shang

**Affiliations:** 1National Engineering Research Center of Edible Fungi, Institute of Edible Fungi, Shanghai Academy of Agricultural Sciences, Shanghai 201403, China; 2College of Marine Resources and Environment, Hebei Normal University of Science & Technology, Qinghuangdao 066004, China

**Keywords:** *Flammulina filiformis*, CRISPR/Cas9, genomic editing, RNPs, *pyrG*

## Abstract

CRISPR/Cas9 systems were established in some edible fungi based on in vivo expressed Cas9 and guide RNA. Compared with those systems, the in vitro assembled Cas9 and sgRNA ribonucleoprotein complexes (RNPs) have more advantages, but only a few examples were reported, and the editing efficiency is relatively low. In this study, we developed and optimized a CRISPR/Cas9 genome-editing method based on in vitro assembled ribonucleoprotein complexes in the mushroom *Flammulina filiformis*. The surfactant Triton X-100 played a critical role in the optimal method, and the targeting efficiency of the genomic editing reached 100% on a selective medium containing 5-FOA. This study is the first to use an RNP complex delivery to establish a CRISPR/Cas9 genome-editing system in *F. filiformis*. Moreover, compared with other methods, this method avoids the use of any foreign DNA, thus saving time and labor when it comes to plasmid construction.

## 1. Introduction

*Flammulina filiformis* from East Asia (previously referred to as *F. velutipes* or *F. velutipes* var. *filiformis*) [1] is a commercially valuable and edible fungus. In recent years, with the development of the *F. filiformis* industry and increased market demand, generating cultivars with high-yield and improved quality has caused important production issues [2]. However, the lack of efficient genetic engineering tools makes it difficult to improve the physiological characteristics of this species [3]. Therefore, the development of new strategic approaches, such as genome editing, are being used to overcome this hurdle [4,5,6].

The CRISPR/Cas9 genome-editing system is a revolutionary technology and a powerful tool for precision molecular breeding [7]. A typical system comprises nuclease (Cas9), mature CRISPR RNA (crRNA), and trans-activating crRNA (tracrRNA). The crRNA can combine with the tracrRNA to generate a single-guide RNA (sgRNA) [8,9], which can effectively induce the Cas9 nuclease to cleave the target sequences. DNA double-strand breaks (DSBs) are formed when the sequence is cleaved. Then, the genomic DNA initiates the repair process. In eukaryotes, there are two DNA self-repair mechanisms: the non-homologous end-joining (NHEJ) and homologous directed repair (HDR) pathways. The NHEJ, as the dominant repair pathway, can lead to genomic alteration by causing random insertion, deletion, or replacement at DSB locations [10,11,12].

CRISPR/Cas9 is now becoming the standard methodology for improving genome-editing efficiency in fungi. Typically, the implementation of CRISPR/Cas9 systems in fungi is based on in vivo expression of Cas9 and sgRNA. Although plasmid construction is becoming straightforward, it takes time for plasmid propagation and extraction. In addition, the most notable merit is that the RNP-based CRISPR-Cas9 is a fast, easy, and accurate strategy in gene editing while avoiding transgenes in many organisms [13]. Thus, implementing Cas9 and the sgRNA components as an in vitro assembled ribonucleoprotein (RNP) complex in transformation may be a viable alternative.

Genetic manipulation of basidiomycete mushrooms is challenging [14]. To facilitate early phenotypic screening of mutants, most studies attempting to establish CRISPR/Cas9 systems have chosen target genes that encode clear morphological phenotypes or physiological properties for editing, such as *pyrG* (encoding orotidine 5′-phosphate decarboxylase) in *Pleurotus eryngii* [15] and *ura3* (syn. *pyrG*) in *Ganoderma lucidum* [16]. As *pyrG* and *ura3* have negative selection effects, which can convert 5-fluorooric acid (5-FOA) into the toxic substance 5-fluorouridine, *pyrG/ura3* mutants survive, and the wild-type (WT) die on a medium supplied with uracil and 5-FOA [9].

Although there have been some reports of the establishment of gene editing systems for *F. filiformis* in recent years [17,18,19], these technologies are all based on genetic modification (GM). However, public attitudes towards GM-based agricultural products are still conservative. Therefore, the establishment of editing technology independent from GM or exogenous DNA is beneficial to evade regulation of policy. Moreover, the reported editing efficiency is still very low [2]. In this study, we used an RNP delivery strategy to develop a CRISPR/Cas9 transformation method in *F. filiformis*, in which the editing efficiency on the *pyrG* gene was 100%.

## 2. Materials and Methods

### 2.1. Strains and Culture Conditions

*Flammulina filiformis* homokaryon strain Dan3, used in this study, was stored at the Shanghai Key Laboratory of Agricultural Genetics and Breeding. *Flammulina filiformis* was cultured on PDA medium (200 g/L potato starch, 20.0 g/L dextrose, and 20.0 g/L agar) at 25 °C. The uracil auxotrophic mutants were grown on PDA containing 100 µg/mL uracil (Sangon Biotech, Shanghai, China).

### 2.2. Screening for Optimum Concentration of Triton X-100 Reagent

Different gradients (0, 0.005%, 0.01%, 0.15%, 0.2%, and 0.3% [*w*/*v*]) of Triton X-100 were prepared to screen its optimum concentration. Triton X-100 is a chemical reagent that can improve cell membrane permeability to ensure RNPs cross the fungal cell membrane and nuclear membrane successfully [20,21].

### 2.3. Preparingation of sgRNA

The protospacer sequence was designed and analyzed online (https://www.idtdna.com/site/order/designtool/index/CRISPR_PREDESIGN, accessed on 12 September 2020). The crRNA containing the 20 bp protospacer sequence for *pyrG* was chemically synthesized together with a 36 bp consensus sequence provided by Integrated DNA Technologies (Coralville, IA, USA). The equimolar concentrations of crRNA and tracrRNA (purchased from IDT, Coralville, IA, USA) were mixed in equal proportions and annealed at 95 °C for 5 min and then left at room temperature (5 °C/min to 25 °C) to form sgRNA. The integrity of the hybridized products was visualized using agarose gel electrophoresis. The sequences of crRNA and tracrRNA are shown in Table 1.

### 2.4. Preparation of RNP Complex

The RNP complex was assembled in a 33 µL volume reaction containing a mixture of 1.6 µL of commercial Cas9 nuclease (62 μM, purchased from IDT), 8.3 µL of sgRNA (10 μM), 3.3 µL of 10 × Cas9 nuclease reaction buffer (20 mM HEPES, 100 mM NaCl, 5 mM MgCl2, 0.1 mM EDTA; pH 6.5), and 19.8 µL of RNase free water. The mixture was incubated for 20 min at room temperature (18–25 °C) to form RNPs.

### 2.5. In Vitro Cas9 Cleavage Assay

To determine the activity of sgRNA, an in vitro cleavage assay was performed. The primers FfpyrG-1F/1R and FfpyrG-4F/4R (Table 2) were used to amplify the *pyrG* fragments containing the target site. One microliter of RNP complex (3000 nM), 2 μL of PCR-amplified fragment (100 ng/μL), 1 μL of 10 × Cas9 nuclease reaction buffer, and 6 µL of RNase-free water were combined and mixed. The mixture was incubated for 1.5 h at 37 °C. Then, 1 µL of protease K (20 mg/mL) was added to terminate the reaction. Finally, the mixture was visualized using 2% agarose gel electrophoresis.

### 2.6. Preparation of Protoplasts

Protoplast preparation and transformation were performed as described previously [16] with minor modifications. The mycelia were collected, washed with 0.6 M mannitol (Sangon Biotech), and digested with 2% (*w*/*v*) lywallzyme (Guangdong Institute of Microbiology, Guangzhou, China) for 90 min. After filtering off the insufficiently digested mycelia, the protoplasts were re-suspended in MTC buffer (0.6 M mannitol, 100 mM CaCl_2_, and 100 mM Tris-HCl; pH 7.5) and adjusted to a range of concentrations: 10^4^, 10^5^, 10^6^, and 10^7^ cells mL^−1^.

### 2.7. PEG-Mediated Transformation of Protoplasts

The resuspended protoplasts were transformed with RNP complex (0, 90, 170, 250, 300, or 400 nM), 1 µL Triton X-100 [0.01% (*w*/*v*) final concentration in transformation reaction], 10 μL 10 × Cas9 nuclease reaction buffer, 31.5 μL 2 × MTC buffer, and 12.5 μL PTC buffer [60% polyethylene glycol (PEG) 4000, 100 mM CaCl2, and 10 mM Tris-HCl; pH 7.5]. The mixture was chilled on ice for 20 min and then 500 μL PTC buffer was added and incubated for another 70 min at 20 °C. Then, 1 mL MTC buffer and 2 mL resuscitation medium PDMU (200 g/L potato starch, 20.0 g/L dextrose, 109.3 g/L mannitol, and 100 µg/mL uracil) were added, and the aliquot was incubated for 24 h at 20 °C. Finally, 3.6 mL low melting top-agar medium PDLMUF (200 g/L potato starch, 20.0 g/L dextrose, 20.0 g/L low melting-point agarose, 109.3 g/L mannitol, 100 µg/mL uracil, and 0.1 mg/mL 5-FOA) was added to the aliquot and poured onto the bottom-medium PDAMUF (200 g/L potato starch, 20.0 g/L dextrose, 20.0 g/L agarose, 109.3 g/L mannitol, 100 µg/mL uracil, and 0.1 mg/mL 5-FOA) selective regeneration medium.

### 2.8. Screening and Verification of Transformants

At 11 d after the PEG-mediated protoplast transformation, all transformants present on PDAMUF selective regeneration medium were transferred to PDAUF medium (200 g/L potato starch, 20.0 g/L dextrose, 20.0 g/L agarose, 100 µg/mL uracil, and 0.1 mg/mL 5-FOA). After 10 d of culture, all isolates were examined using diagnostic PCR or sequencing analysis utilizing mycelia as the template. With either the primers FfpyrG-1F/1R or FfpyrG-4F/4R, PCR amplification of the *pyrG* fragments from all the transformants that contained the target sites was compared with the WT *pyrG* sequence. Sanger sequencing was used to determine whether indels occurred at, or near, the expected sites. Randomly selected transformants were transferred to PDA plates containing uracil and subcultured 2–4 times to ensure hereditary stability.

## 3. Results

### 3.1. Effects of Different Concentrations of Triton X-100 on Protoplast Regeneration

The surfactant Triton X-100 can be used to improve cell membrane permeability and ensure that RNPs successfully cross the fungal cell membrane and nuclear membrane during PEG-mediated protoplast transformation [20,21,22,23]. However, overloading Triton X-100 causes cell death due to the complete destruction of the cell membrane. Therefore, we first analyzed the optimal concentration of Triton X-100 for *F. filiformis*. The results showed that, as the concentration of Triton X-100 increased, the number of regenerated protoplasts showed an obvious reduction. Although the number of protoplasts (123 colonies in Figure 1C) obtained at the concentration of 0.01% Triton X-100 has decreased, it is still sufficient for the next step, prompting our choice of 0.01% as the optimum concentration (Figure 1).

### 3.2. In Vitro Cas9 Cleavage Assay

In many reported fungal CRISPR systems, *pryG* is commonly employed as a bidirectional selective marker to avoid the use of antibiotic screening markers. To verify whether the pyrG target site could be recognized and cleaved by Cas9 endonuclease under the guidance of designed sgRNA, we performed an in vitro cleavage validation experiment (Figure 2A). Using genomic DNA of *F. filiformis* as the template, the pyrG fragments containing target regions with sizes of 578 bp and 902 bp were amplified through the primers FfpyrG-1F/FfpyrG-1R and FfpyrG-4F/FfpyrG-4R, respectively. Theoretically, cleavage of the 578 bp fragment by the RNP complex would yield two small bands of 367 bp and 210 bp, while cleavage of the 902 bp fragment would yield two small bands of 427 bp and 475 bp (Figure 2B). The result of the agarose gel electrophoresis showed that the crRNA and tracrRNA annealed in vitro to form a stable sgRNA, with complete structure and no degradation (Figure 2C). As expected, the 578 bp and 902 bp PCR fragments were almost completely digested, and two respective small bands appeared when the RNP complex was present in the reaction system (Figure 2D). The results were consistent with our expectations and indicated that the targeting efficiency of the sgRNA was sufficient to be used in subsequent experiments.

### 3.3. Optimization of Polyethylene Glycol (PEG)-Mediated Protoplast Transformation of RNPs in F. filiformis

The amounts of RNPs and protoplasts are pivotal factors that restrict the editing efficiency in an RNP-based editing strategy. To determine the effect of protoplast concentration on the targeting efficiency, four concentration gradients were set (0, 10^4^, 10^5^, 10^6^, and 10^7^ cells mL^−1^). After 11 days of incubation on a regeneration medium (containing 100 µg/mL uracil and 0.1 mg/L 5-FOA), the resulting number of colony-forming units (CFUs) was determined, and the corresponding efficiency was analyzed by PCR and sequencing with the FfpyrG-1F and FfpyrG-1R primers (Table 3, Figure 3 and Appendix A). Five transformants were obtained from transformations with protoplasts at concentrations of 10^7^ cells mL^−1^ when 0.01% Triton X-100 was added. However, no transformants were obtained in the control group without Triton X-100 (Table 3). This result indicates that Triton X-100 promoted the passage of RNP through the cell membrane, and gene editing ensued. Conversely, we also assessed the edited transformants grown on non-selective media. Although only 4.07% of the colonies were edited, this result demonstrated that it was feasible to obtain correct transformants independent of selective marker or transgenic system.

To determine the effect of the RNP concentration on the targeting efficiency, RNPs at a range of concentrations (0, 90, 170, 250, 300, and 400 nM) were transformed into *F. filiformis* protoplasts (10^7^ cells mL^−1^) and an addition of 0.01% Triton X-100. No transformants were obtained without RNPs or with RNPs at concentrations ≤ 170 nM. When the RNP concentration was 300 nM, the resulting number of CFUs was at a maximum (Table 4 and Figure 4). Therefore, 300 nM is the optimal RNP concentration (Figure 5).

The *pyrG* genes of 16 *F. filiformis* transformants were sequenced, all of which exhibited expected mutations in the vicinity of the cleavage site (Figure 3, Figure 4 and Appendix A). Sequence analysis of the transformants with insertions revealed that the inserted fragments, 115 bp and 47 bp, were consistent with the genome sequences of JAJAKW010000006.1 and JAJAKW010000005.1, respectively [Fv01-10_genome GenBank assembly (GCA_022345005.1)]. The other two inserted fragments of transformant 10^6^-2 and 10^7^-3 were not able to align with the genome of *F. filirormis*.

### 3.4. Comparative Analysis of RNP-Directed Mutants and Wild-Type Strain

As a genetic manipulation toolbox, the application of such a tool should be avoided to cause phenotypic changes. To ensure that the transient introduction of RNPs would not disrupt the expression and function of other genes, the morphology and important phenotypic traits—such as growth and cellulase activity—of the selected *F. filiformis* transformants, which had undergone *pyrG* mutagenesis, were compared with those of the parental strains. The mycelial growths of five randomly selected RNP-directed mutants were observed on potato dextrose agar (PDA) medium, with or without uridine, after 7 d of incubation. On PDA plates without uridine, only the prototrophic strain Dan3 grew normally; the other five mutants showed defects in mycelial growth. By contrast, the growth of the mutants on PDA plates containing uridine was not affected (Figure 6A). No significant differences in cellulase and laccase activity were observed among the RNP-directed mutants and the wild-type Dan3 strain (Figure 6B,C).

## 4. Discussion

CRISPR/Cas9 technology has dramatically improved the efficiency of genome editing [24,25]. Determining appropriate target sites is helpful for evaluating editing efficiency [26]. As pyrG has a negative selection effect, growth inhibition can be easily observed in PDA plates containing 5-FOA (cells with wild-type pyrG converting 5-FOA into the toxic substance 5′ fluorouridine monophosphate). In this study, we produced mutations using RNP complex transformation. The indels exhibited classic mutations in the vicinity of the cleavage site. In contrast to the plasmid method, the RNP strategy is completely free of foreign DNA and can save time and labor by eliminating the tedious steps involved in plasmid construction. In the future, endogenous *pyrG* can be used to develop functional gene knockouts that do not require antibiotic markers, and the *pyrG* marker recycling system can be employed to implement multiple rounds of gene editing [27].

There are three common delivery strategies in the CRISPR-Cas9 genome-editing system: Cas9 nuclease in vivo and sgRNA in vitro (Cas9-expressing chassis with sgRNA in vitro), both Cas9 and sgRNA in vivo (all-in-one plasmid harboring Cas9-expressing cassette and sgRNA cassette), and both Cas9 and sgRNA in vitro (RNP complex) [28]. In this context, Moon et al. successfully disrupted the LeA1 locus of *Lentinula edodes* by delivering a plasmid containing the LeU6 and LeGPD promoters to express the Cas9 protein [29]. Wang et al. established a CRISPR/Cas9 genome-editing system in *G. lucidum* based on a plasmid delivery strategy, but the editing efficiency was low [14]. Boontawon et al. established an efficient CRISPR/Cas9-assisted genome-editing system based on plasmid harboring expression cassettes of Cas9 and different single-guide RNAs in *P. ostreatus* [30]. Liu et al. reported a CRISPR/Cas9 genome-editing system based on a plasmid delivery strategy in *F. filiformis* but only two mutants were obtained in their study [2]. The transformation efficiency of the plasmid method is relatively low, and there are many false positives. The RNPs strategy involves the direct delivery of an in vitro Cas9/sgRNA complex. In basidiomycetes, there was a successful transformation of RNPs, such as in *Schizophyllum commune* [31] and *P. ostreatus* [32], but only a few mutants were obtained. In our study, the Cas9/sgRNA complex was used in *F. filiformis* for direct delivery into the protoplasts through PEG-mediated transformation. In this case, the targeting efficiency of the genomic editing was 100% when the mutants were selected on a medium containing 5-FOA. The addition of the surfactant Triton X-100 may be the key to this high efficiency. Triton X-100 can improve cell membrane permeability and ensure that the RNPs cross the fungal cell membrane and nuclear membrane successfully during PEG-mediated protoplast transformation [20,21,22]. However, protoplasts cannot be regenerated owing to the complete disintegration of the cell membrane structure caused by excessive Triton X-100 (>0.2%), therefore, milder surfactants than Triton X-100 may be an alternative for highly efficient delivery for RNPs.

Notably, the plasmid method used with *F. filiformis* required up to 28 d for the transformants to grow after PEG-mediated transformation [2]. However, in our RNP complex method, transformant growth occurred after only 11 d. The reason for this difference is unknown. It is possible that the unwanted genomic integrations of DNA constructs expressing Cas9 and sgRNA affect the growth rate of *F. filiformis*.

In this study, the morphology and important phenotypic traits showed no significant differences between the RNP-directed mutants and the parental strain *F. filiformis* Dan3 (Figure 6). However, when mutants of *Trichoderma reesei* were obtained based on Cas9 plasmids, cellulase activities were indeed affected [23]. The authors considered that in vivo expression of Cas9 might impact ordinary physiological and biochemical processes in the transformants, which could be related to the endogenous promoter used to express the Cas9 protein. The RNP complex delivery strategy can avoid this shortcoming.

Sequence analysis of the transformants with insertions revealed that the 115 bp and 47 bp fragments were consistent with the genome sequences of JA-JAKW010000006.1 and JAJAKW010000005.1, respectively, in the *F. filiformis* genome. A large fragment insertion also occurred in the filamentous fungi *G. lucidum* [16] and *Aspergillus nidulans* [33]. Some large insertions were also identical to other genomic loci but the mechanism of capturing fragments from the whole genome to assist NHEJ remains unclear.

In conclusion, this report demonstrates a successful CRISPR/Cas9 genome-editing system through direct delivery of an RNP complex into the cultivated mushroom *F. filiformis*. This system is free from genomic integration of ectopic sequences. In the future, we will use uridine auxotrophic mutants as parental strains to study other functional genes using the HDR mechanism with donor DNA.

## Figures and Tables

**Figure 1 jof-08-01000-f001:**
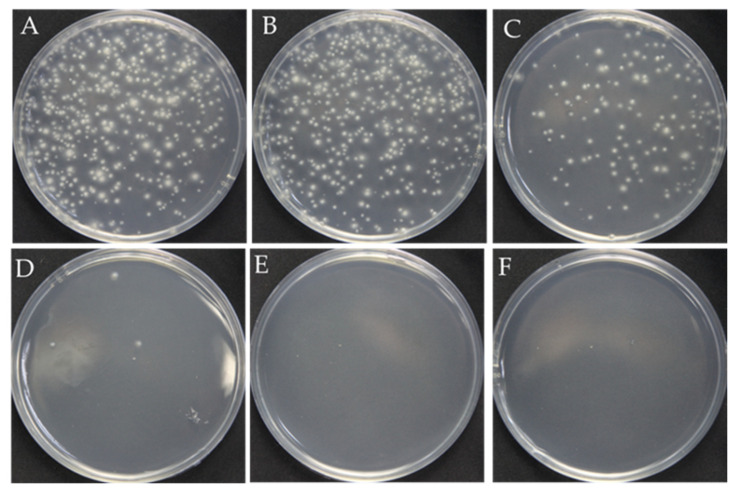
Effect of the Triton X-100 on the protoplast regeneration. (**A**–**F**): The different concentrations of Triton X-100: 0, 0.005%, 0.01%, 0.15%, 0.2%, 0.3%, respectively.

**Figure 2 jof-08-01000-f002:**
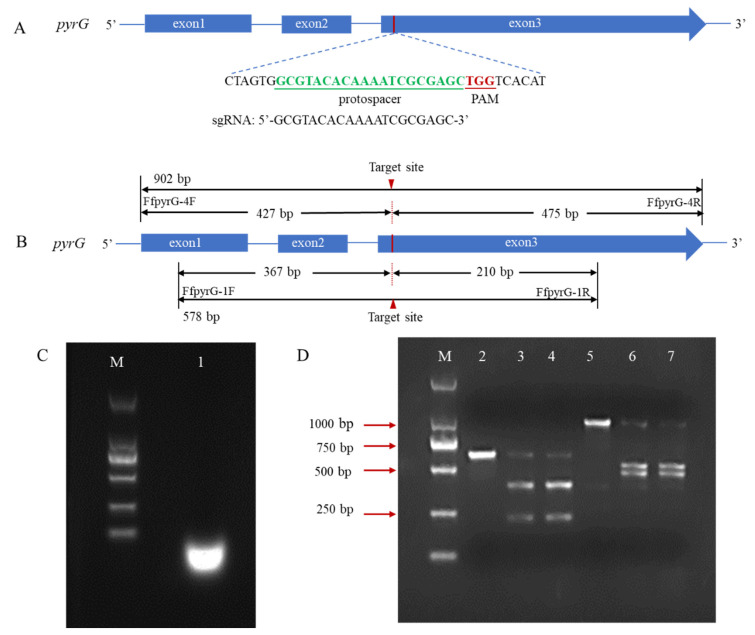
Schematic illustration of sequence information and in vitro Cas9 cleavage assay. (**A**) Sequences of the sgRNA targeting pyrG are shown in green font, which were located at exon3. Sequence direction was 5′-3′ as shown. Schematic representations of exons are drawn to scale. The protospacer adjacent motif (PAM) is shown in red. (**B**) Fragments required for the in vitro Cas9 cleavage assay were amplified with primers FfpyrG-1F/FfpyrG-1R and FfpyrG-4F/FfpyrG-4R. The red triangle represents the cleavage site of Cas9. The annealed products of crRNA and tracrRNA (**C**) and cleavage assay of Cas9 nuclease in vitro were visualized using agarose gel electrophoresis (**D**) M, DL 2000 marker; 1, sgRNA products; 2, pyrG-578 bp without RNPs; 3–4, pyrG-578 bp with RNPs; 5, pyrG-902 bp without RNPs; 6–7, pyrG-902 bp with RNPs.

**Figure 3 jof-08-01000-f003:**
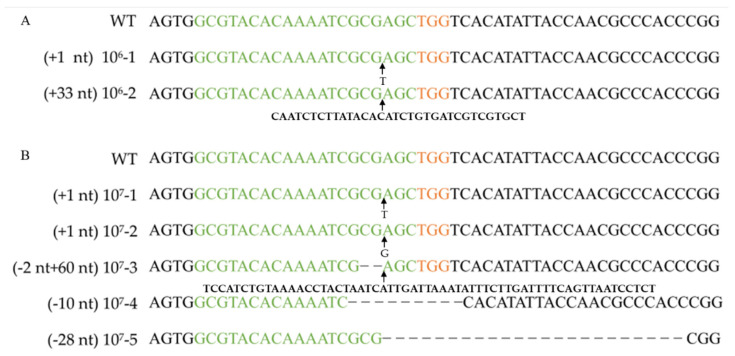
Verifying *pyrG* mutation after RNP-based gene disruption using different protoplast concentrations. Alignment of *pyrG* sequences and corresponding mutants obtained by transformation with 10^6^ cells mL^−1^ protoplasts (**A**) and 10^7^ cells mL^−1^ protoplasts (**B**). WT: wild-type strain Dan3. Concentrations 10^6^-1, 10^6^-2, 10^7^-1~10^7^-5: mutants generated by transformation of RNPs. Green marked represents spacer sequences; orange marked represents PAM.

**Figure 4 jof-08-01000-f004:**
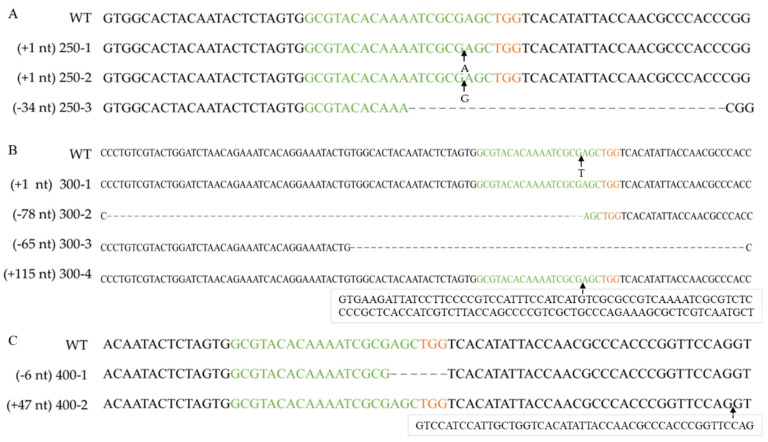
Verifying *pyrG* mutation after RNP-based gene disruption using different RNP concentrations. Alignment of *pyrG* sequences and corresponding mutants obtained by transformation with 250 nM RNPs (**A**), 300 nM RNPs (**B**) or 400 nM RNPs (**C**). WT: wild-type strain Dan3. Concentrations 250-1~250-3, 300-1~300-4, 400-1~400-2: mutants generated by transformation of RNPs. Green marked represents spacer sequences; orange marked represents PAM.

**Figure 5 jof-08-01000-f005:**
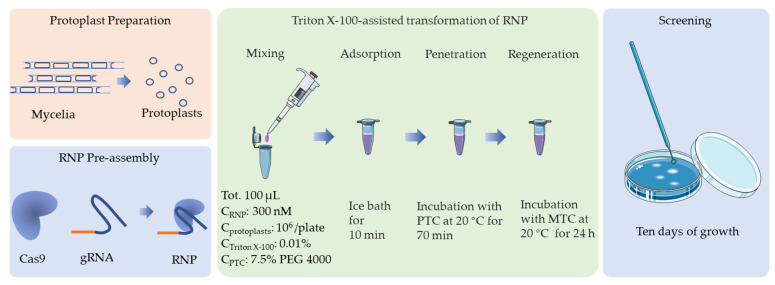
Schematic representation of the optimal protocol for CRISPR/Cas9 transformation of *F. filiformis*, which is based on an in vitro assembled RNP complex.

**Figure 6 jof-08-01000-f006:**
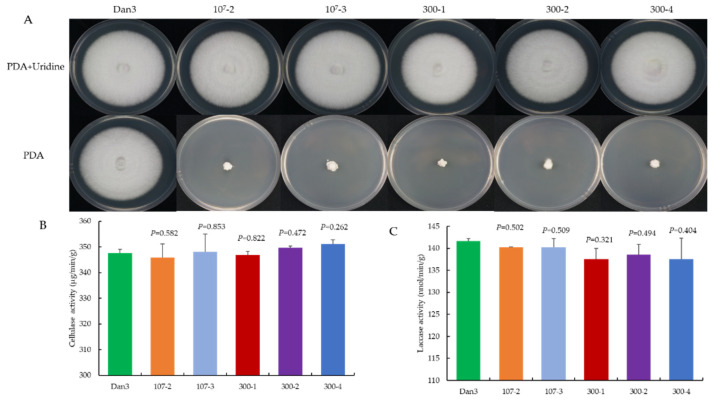
Comparative analysis of RNP-directed mutants and their parental strains. (**A**) Phenotype of Dan3 and its *pyrG* mutants on PDA plates (with/without uridine). Dan3: wild-type. 10^7^-2, 10^7^-3, 300-1, 300-2, 300-4: *pyrG* mutants edited with RNP. (**B**) Cellulase activities of Dan3 and corresponding *pyrG* mutants. (**C**) Laccase activities of Dan3 and corresponding *pyrG* mutants. Error bars indicate the standard deviation of three replicates. All *p*-values were obtained using two-tailed t-tests in Microsoft Excel.

**Table 1 jof-08-01000-t001:** RNA sequence information.

Name	Sequence (5′ to 3′)
crRNA (56 bp)	GCGTACACAAAATCGCGAGCCUCCUUCACCUCCUCUCAUCGUUUUAGAGCUAUGCU ^1^
tracrRNA (67 bp)	AAAUAGCAAGUUAAAAUAAGGCUAGUCCGUUAUCAACUUGAAAAAGUGGCACCGAGUCGGUGCUUUU

^1^ The underline indicates the protospacer sequences.

**Table 2 jof-08-01000-t002:** List of oligonucleotides used in this study.

Name	Fragment Length (bp)	Sequence (5′ to 3′)
*pyrG*-1F	578	GAGACTATGGAACGCAAAA
*pyrG*-1R		CCTCTGAGCGATGAAGC
*pyrG*-4F	902	ATGCAGTCCTACGCCGCTCG
*pyrG*-4R		TCATGCTGTTCTCTCCAAGT

**Table 3 jof-08-01000-t003:** Effect of protoplast concentration on targeting efficiency.

Protoplasts (Cell mL^−1^)	0.01% Triton X-100		No Triton X-100	
	CFUs	Positivity Rate (%)	CFUs	Positivity Rate (%)
3.5 × 10^4^	0	-	0	-
3.5 × 10^5^	0	-	0	-
3.5 × 10^6^	2	100	0	-
3.5 × 10^7^	5	100	0	-

**Table 4 jof-08-01000-t004:** Effect of the RNPs concentration on targeting efficiency.

RNPs (nM)	CFUs	Positivity Rate (%)
0	0	-
90	0	-
170	0	-
250	3	100
300	4	100
400	2	100

## Data Availability

Not applicable.

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
