# Peer review of "A Simple and Efficient CRISPR/Cas9 System Using a Ribonucleoprotein Method for Flammulina filiformis"

_jof, 2022, doi:10.3390/jof8101000_

Round 1

Reviewer 1 Report

The authors develop a method a deliver in vitro assembled Cas9 and sgRNA ribonucleoprotein complexes to the muchroom Flammulina filiformis, and achieved good editing result. The experiments were well-designed, results were clearly-presented, I recommend publishing in current form.  

Author Response

Thanks for your positive comments.

Reviewer 2 Report

In their report „A simple and efficient CRISPR/Cas9 system using a ribonucle-2 oprotein method for Flammulina filiformis” Liu, Cui et al. developed an mRNP-based CrispR approach to generate mutants of the pyrG gene in the edible fungus F. filiformis. They optimize the transformation procedure by adding the nonionic detergent Triton X-100 to fascilitate uptake of in vitro assembled RNPs. Using this procedure they were able to obtain several pyrG mutants able to grow in the presence of 5-FOA. Characterization of the sequence allowed them to validate that mutations are likely to result from CrispR-mediated cleavage and subsequent non-homologous repair mechanisms. Although the paper merits publication in principle as it shows the development of an interesting tool to manipulate the genome of an edible fungus, I have a few comments, which should be addressed prior to publication.

1.     In Figure 2 the authors show the depiction of an in vitro cleavage assay of pyrG derived DNA fragments. It is puzzling that the samples have been mixed up prior to loading. Lane 2 shows the control fragment for the result depicted in lane 5 and lane 4 the control fragment for the result depicted in lane 3. This is confusing and the picture should be replaced.

2.     The authors make the strong statement that their editing rate is 100%. This may only be a consequence of the specific gene they target and the way they select. Transformants are recovered on 5-FOA containing medium, which only allows growth of pyr mutants. This experimental set-up is not clear from the abstract. Thus, readers might get the unjustified impression that the editing rate is 100% without any preselection being applied. I suggest that this issue needs to be clarified already in the abstract section of the paper.

3.     Have the authors tested, which percentage of colonies showed growth on FOA containing medium compared to non-selective medium? This information would be important to judge the value of their system – especially because they advertise the tool as a way to circumvent a “transgenic” approach.

Minor points:

Line 60: F. filiformi should be F. filiformis.

Line 94: RNase free … water?

Line 147: next process should be replaced: the next step?

Line 157: F. filiformis should be in italics.

Line 201: “classic mutations” should be replaced by “expected mutations”.

Line 206: F. filiformi should be F. filiformis.

Lines 205-206: Have the authors tried to identify these sequences via “Blast” searches optimized for short sequences?

Reviewer 3 Report

The manuscript entitled "A simple and efficient CRISPR/Cas9 system using a ribonucle- 2 oprotein method for Flammulina filiformis" can be accepted for publication in its current form. My only concern is figure 5. If the author can get a graphical designer to help with the figure?

Author Response

Thanks for your positive comments. Figure 5 has been redesigned in the revised manuscript.

Reviewer 4 Report

Liu et al. reported the use of CRISPR-Cas9 RNP complex for genome editing of Flammulina filiformis. The authors optimized the transformation of CRISPR-Cas9 RNP by titrating the reagents and realized gene editing. It is interesting to have a simple and efficient genome editing approach for future use on Flammulina filiformis, but there are some concerns that preclude the publication of this study in the Journal of Fungi.

Major comments

The authors emphasized in the study that the apparent advantage of RNP is less work effort compared to plasmid delivery. However, plasmid construction nowadays is getting much easier and this argument is not strong enough for the choice of RNP delivery. Moreover, a similar RNP transformation protocol has been described in other Fungi species, which somewhat reduces the novelty of this current study. To strengthen the rationale of RNP use, one should demonstrate the other traits of RNP compared to plasmids in Flammulina filiformis. For example, CRISPR-Cas9 RNP is more efficient and more specific in gene editing in mammalian cells and it would great to show if similar phenomena could be observed in Flammulina filiformis.

Minor comments

1. The possible impact on Flammulina filiformis should be discussed. Are there other substitutes less toxic available besides Triton X-100?

2. Figure 2D is ugly: there are too many marker lanes; lane 2 should be next to 5; lane 3 should be next to 4. Quantification would be ideally quantified.

3. Figure 3 should be made clearer. The sequence information in between belongs to Figure 2B, not 2A. It is very difficult for the reads to pinpoint the sequence changes in the chromatograms, thus the sequence changes in chromatograms have to be made clear if they are to be displayed. The same applies to Figure 4.

4. The authors compared the phenotypes of wild-type and mutants. It would be more convincing that genotypes, such as the top gRNA off-target sites for the mutants.

5. In the method section, the annealing of tracerRNA and crRNA is done not only by heating at 95 °C for 5 min, but also followed by a cooling down step. 

Round 2

Reviewer 2 Report

Dear authors, all my concerns have been addressed. Many thanks for this. 

Author Response

Thanks for your positive comments.

Reviewer 4 Report

1.

Page 2 line 49> please correct the sentence: "...CRISPR-Cas9 exhibited faster, easier, and more accurate what? in gene editing...";

 line 49> regarding "the science is still 53 incomplete. " What science means?

Similarly, please check all through the manuscript to make the manuscript more readable!

2.

Figure 3 and 4> It is very rare and confusing to present wt sequence containing dash line. If the insertion sequence is large, one could simply add it in another line to make it clear to readers.  
